# Graph-based Topology Reasoning for Driving Scenes

## Abstract

Understanding the road structure is essential for achieving autonomous driving. This intricate topic contains two fundamental components - the interconnections between lanes and the associations between lanes and traffic elements (e.g., traffic lights), where a comprehensive topology reasoning method is still absent. On one hand, existing map learning techniques face challenges in deriving lane connectivity using segmentation or laneline-based representations; or prior approaches focus on centerline detection while neglecting interaction modeling. On the other hand, the topic of assigning traffic elements to lanes is limited in the image domain, leaving the construction of correspondence between image and 3D views as an unexplored challenge. To address these issues, we present **TopoNet**, an end-to-end topology reasoning network for analyzing driving scenes. To capture the topology of driving environments effectively, we introduce three key designs: (1) an embedding module that integrates semantic knowledge from 2D elements into a unified feature space; (2) a curated scene graph neural network that models relationships and facilitates feature interactions within the network; (3) instead of transmitting messages arbitrarily, a scene knowledge graph is devised to differentiate prior knowledge from various types of the scene topology. We evaluate TopoNet on the challenging scene understanding benchmark, OpenLane-V2, where our approach outperforms all previous works by a great margin across all perceptual and topological metrics. The code will be publicly released.

## 1 Introduction

Imagine an autonomous vehicle navigating towards a complex intersection and planning to go straight ahead: it wonders when choosing the appropriate lane to enter and determining which traffic signal to adhere to. This sophisticated challenge necessitates the agent to not only accurately perceive lane positions, but also understand the topological relationships from sensor inputs. Specifically, the topology in a driving scene includes: (1) the **lane topology graph** comprising centerlines as well as their connectivity, (2) and the **assignment relationships** between lanes and various traffic elements such as traffic lights and road markers. As depicted in Fig. 1, they collectively form a topological structure that furnishes explicit navigation cues essential for downstream tasks like motion prediction and planning (Bansal et al., 2018; Chai et al., 2020).

Conventional driving datasets (Caesar et al., 2020; Wilson et al., 2021) typically incorporate lane topology implicitly within High-Definition (HD) maps, which are primarily designed for data storage but not for neural networks' learning. Various formulations have been proposed to substitute HD maps, such as 2D and 3D laneline detection (Pan et al., 2018; Garnett et al., 2019; Tabelini et al., 2021; Chen et al., 2022), bird's-eye-view (BEV) map element detection through segmentation (Pan et al., 2020; Roddick & Cipolla, 2020; Li et al., 2022a; Hu et al., 2023) and vectorization (Liu et al., 2023a; Liao et al., 2023a;b). To derive lane connectivity, a naïve strategy is to directly average the coordinates of two neighboring lanelines to get lane centerlines, and then construct a lane graph based on the centerline instances. Yet, it requires complicated hand-crafted rules and extensive post-processings. An alternative approach is to supervise perception networks with relationship labels. Recent studies (Can et al., 2021; 2022a) employ a Transformer-based architecture for lane instances prediction and an additional Multi-layer Perceptron (MLP) to learn connectivity. Nevertheless, they suffer from extracting valuable information without explicit modeling of relationships.

Moreover, the problem of assigning relationships between traffic elements and lanes based on sensor inputs remains largely unexplored. Langenberg et al. (2019) attempted to associate the ground truth representations

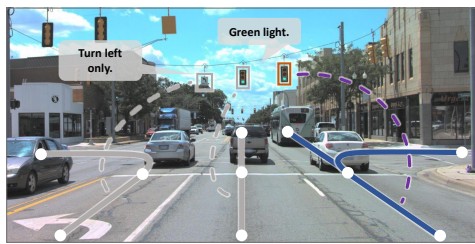 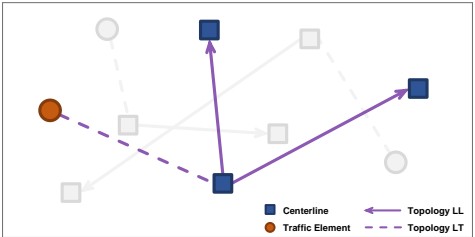

Figure 1: **Topology relationships of a driving scene.** While approaching an intersection, an autonomous vehicle has to reason about the correct lane and traffic information for subsequent navigation. We advocate, and present TopoNet, to directly achieve topology understanding on the heterogeneous graph. "Topology LL" and "Topology LT" represent the relationship among lane centerlines and the associations between lane centerlines and traffic elements, respectively.

of lanelines and traffic lights in the image domain (specifically the perspective view, PV). However, integrating traffic elements and lanes within a heterogenous graph (Fig. 1) presents a distinct set of challenges. One key obstacle is that traffic elements are typically described as bounding boxes in PV, whereas lanes are depicted as curves in 3D or BEV space. Besides, spatial locations remain less important for traffic elements as their semantic meanings are essential, but positional clues of lanes are crucial for autonomous vehicles.

To address these issues, we present a Topology Reasoning Network (**TopoNet**), which predicts the driving scene topology in an end-to-end manner. As an attempt to reason about scene topology in a single network, TopoNet comprises a shared feature extractor, and two detection branches for traffic elements and centerlines respectively. Motivated by the Transformer-based detection algorithms (Carion et al., 2020; Zhu et al., 2020), we employ instance queries to extract local features via the deformable attention mechanism, which confines the attention region and accelerates convergence. Since the clues for locating a specific centerline instance might lie in its neighboring elements and related traffic elements' features, a Scene Graph Neural Network (SGNN) is devised to facilitate message passing among instance-level embeddings. Furthermore, we propose a scene knowledge graph to capture prior topological knowledge from entities of different types. Specifically, a series of GNNs are developed based on the categories of traffic elements and the centerline connectivity relationships (i.e., predecessor, ego, successor). Updated queries are decoded to yield perception results and topology relationships. With the proposed designs, we deploy TopoNet on the large-scale topology reasoning benchmark, OpenLane-V2 (Wang et al., 2023). TopoNet outperforms state-of-the-art approaches by 15-84% in centerline perception, 38-270% in topology reasoning tasks, and 37% in terms of the overall perception and reasoning metric. Ablations are conducted to verify the effectiveness of our framework.

## 2 Related Work

### 2.1 Lane Graph Learning

Lane Graph Learning has gained significant attention due to its pivotal role in autonomous driving. Prior works investigate building road graphs (He et al., 2020; Bandara et al., 2022) or more densely structured lane graphs (Homayounfar et al., 2019; Zürn et al., 2021; He & Balakrishnan, 2022; Büchner et al., 2023) from aerial images. However, roads in aerial images are often occluded by trees and buildings, resulting in significant inaccuracies. Recently, there has been a growing focus on producing lane graphs from sensors mounted on vehicles. STSU (Can et al., 2021) proposes a DETR-like network to detect centerlines and then derive their connectivity by a subsequent MLP module. Building upon STSU, Can et al. (2022a) introduce minimal cycle queries to ensure the proper order of overlapping lines. CenterLineDet (Xu et al., 2023) treats centerlines as vertices and designs a graph-updating model trained by imitation learning. LaneGAP (Liao et al., 2024) proposes a path-wise modeling approach to represent the lane graph. It is also worth noticing that Tesla proposes the concept of the "language of lanes" to depict the lane graph as a sentence (Tesla, 2022). The attention-based model recursively predicts lane tokens and their connectivity. In this work, we focus on explicitly modeling the centerline connectivity within the network to enhance feature learning and incorporating traffic elements in the construction of a full driving scene graph.

## 2.2 HD Map Perception

With the trending of BEV perception (Philion & Fidler, 2020; Li et al., 2022b; Zhou & Krähenbühl, 2022; Liao et al., 2023b), recent works focus on learning HD maps with segmentation and vectorized methods. Map segmentation works predict the semantic interpretation of each BEV grid, such as lanelines and pedestrian crossings. They differentiate from each other primarily in the perspective view to BEV transformation module, i.e., IPM-based (Xie et al., 2022; Can et al., 2022b), depth-based (Hu et al., 2022; Liu et al., 2023b), or Transformer-based (Li et al., 2022b; Jiang et al., 2023). Though dense segmentation provides detailed pixel-level information, it falls short in capturing the complex relationships between overlapping elements. Li et al. (2022a) address the problem by grouping and vectorizing the segmented map with sophisticated post-processings. VectorMapNet (Liu et al., 2023a) proposes to directly represent each map element as a sequence of points and uses coarse key points to decode laneline locations sequentially. MapTR (Liao et al., 2023a) further explores unified permutation-based modeling to eliminate ambiguities in point sequence ordering and enhance both performance and efficiency. In fact, vectorization-based methods could be easily adapted for centerline perception by adjusting the supervision since they have enriched the direction information for lanelines. Shin et al. (2023) construct map elements as a graph by initially predicting vertices and then utilizing a GNN module to detect edges. However, its GNN produces all vertex features simultaneously, limiting instance-level interactions. Contrary to them, we leverage instance-wise feature transmission within the GNN, enabling the extraction of significant prediction cues from other elements in the topology graph.

## 2.3 Driving Scene Understanding

The concept of Driving Scene Understanding primarily involves the comprehension of the spatial relationships among elements within outdoor environments, extending beyond mere detection (Tian et al., 2020; Mylavarapu et al., 2020b; Zipfl & Zöllner, 2022; Malawade et al., 2022a). Previous works focus on utilizing the relationships of 2D bounding boxes for motion prediction (Li et al., 2020; Mylavarapu et al., 2020a;b; Fang et al., 2023) and risk assessment (Yu et al., 2021; Malawade et al., 2022b). In the industrial context, Mobileye presents an optimization-based method to construct lane topology and relationships between traffic lights and lanes automatically based on their proprietary data sources (Mobileye, 2022). In the academia, Langenberg et al. (2019) address the traffic light to lane assignment (TL2LA) problem with a convolutional network by taking heterogeneous metadata as additional inputs. In contrast, TopoNet takes only RGB images as inputs and additionally reasons about lane entities' topology. We train and evaluate TopoNet on the large-scale driving scene understanding benchmark, which covers complicated urban scenarios.

## 2.4 Graph Neural Network

Graph Neural Network and its variants, such as graph convolutional network (GCN) (Kipf & Welling, 2017), GraphSAGE (Hamilton et al., 2017), and GAT (Veličković et al., 2018), are widely adopted to aggregate features of vertices and extract information from graph (Scarselli et al., 2008). Witnessing the impressive achievements of GNN in various fields (e.g., recommendation system and video understanding) (Guo & Wang, 2020; Mohamed et al., 2020; Chang et al., 2021; Pradhyumna & Shreya, 2021), researchers in the autonomous driving community try utilizing it to process unstructured data. Weng et al. (2020; 2021) introduce GNN to capture interactions among agents for 3D multi-object tracking. LaneGCN (Liang et al., 2020) constructs a lane graph from HD map, while others (Jia et al., 2022; 2023; Fang et al., 2023) model the relationship of moving agents and lanelines as a graph to improve the trajectory forecasting performance. Inspired by prior works, we design a GNN for the driving scene understanding task to enhance feature interaction and introduce a class-specific knowledge graph to better integrate semantic information.

# 3 TopoNet

## 3.1 Problem Formulation

Given multi-view images, the goal of TopoNet lies in two perspectives - perceiving entities and reasoning their relationships. As an instance-level representation is preferable for topology reasoning, a directed lane centerline (LC) is described by an ordered list of points. We denote it as $v_l = [p_0, ..., p_{n-1}]$, where $p = (x, y, z) \in \mathbb{R}^3$ describes a point's coordinate in 3D space, $p_0$ and $p_{n-1}$ are the starting and ending point.

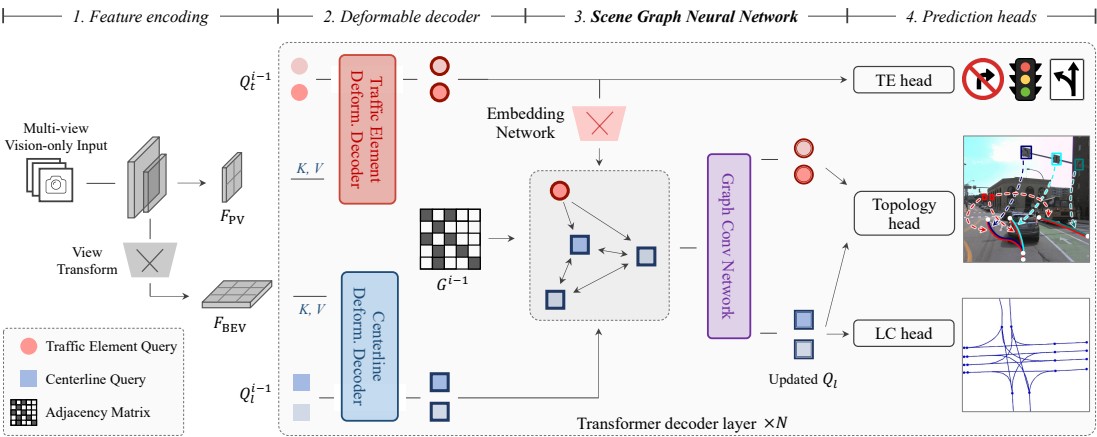

Figure 2: **Systematic diagram of TopoNet**. TopoNet addresses the crucial problem of topology reasoning for driving scenes in an end-to-end fashion. It consists of four stages, with the latter three compacted in a Transformer decoder architecture. TopoNet handles traffic elements and centerlines as two parallel branches at the Deformable decoder. Various types of instance queries (red, blue) then interact, exchange messages, acquire and aggregate prominent knowledge in the proposed Scene Graph Neural Network. The explicit relationship modeling inside the network serves as a favorable scheme for feature learning and topology prediction. In this paper, we abbreviate traffic elements and lane centerlines as "TE" and "LC", respectively.

Traffic elements (TE) are represented as 2D bounding boxes in different classes on the front-view images. All existing lanes $V_l$ and traffic elements $V_t$ within a predefined range are required to be detected.

On the perceived instances, the topology relationships are built. The connectivity of directed lanes establishes a map-like network on which vehicles can drive. We denote the lane graph as $G_{ll} = (V_l, E_{ll})$, where the edge set $E_{ll} \subseteq V_l \times V_l$ is asymmetric. An entry $(i, j)$ in $E_{ll}$ is positive if and only if the ending point of the lane $v_i$ is connected to the starting point of $v_j$. The graph $G_{lt} = (V_l \cup V_t, E_{lt})$ describes the correspondence between lanes and traffic elements. This graph can be interpreted as a bipartite structure, where positive edges only exist between $V_l$ and $V_t$. Given the instance set $V_l$ and $V_t$, the connectivity of predicted graph $G_{ll}$ and $G_{lt}$ is represented by the adjacency matrices $A_{ll}$ and $A_{lt}$, respectively. These matrices are required to be predicted in the task of topology reasoning.

## 3.2 Overview

Fig. 2 illustrates the overall architecture of the proposed **TopoNet**. Given multi-view images, the feature extractor generates multi-scale features, including a front-view feature $F_{\text{PV}}$ and a BEV feature $F_{\text{BEV}}$. Two independent decoders with the same deformable attention mechanism (Zhu et al., 2020) take $F_{\text{PV}}$ and $F_{\text{BEV}}$ to update instance-level embeddings $Q_t$ and $Q_l$, respectively. The proposed **Scene Graph Neural Network (SGNN)** then refines centerline queries $Q_l$ in positional and topological aspects. The decoders and SGNN layers are stacked iteratively for $N$ layers to obtain local and global feature interactions. Task-specific heads employ the refined queries to get final predictions. Next, we elaborate on the proposed SGNN.

## 3.3 Scene Graph Neural Network

A representative embedding (or query) provides ideal instance-wise detection or segmentation results, as discussed in previous perception works (Carion et al., 2020; Wu et al., 2022). However, being discriminative is not enough to recognize correct topology relationships. The reason is that it takes a pair of instance queries as input to determine their relationship, in which feature embeddings are actually not independent. Meanwhile, adopting the local feature aggregation scheme of point-wise queries (Liu et al., 2023a; Liao et al., 2023a) for centerline perception is inadequate. Specifically, a key difference between centerlines and physical map elements is that centerlines naturally encode lane topology and traffic rules, which cannot be inferred from local features alone. Therefore, we aim to simultaneously acquire perception and reasoning results by modeling not only discriminative instance-level representations but also inter-entity relationships.

To this end, we present SGNN, which has several designs and merits compared to previous works. (1) It adopts an embedding network to extract TE knowledge within a unified feature space. (2) It models all entities in a frame as vertices in a graph, and strengthens interconnection among perceived instances to learn their inherent relationships with a graph neural network. (3) Alongside the graph structure, SGNN incorporates prior topology knowledge with a scene knowledge graph.

### 3.3.1 Embedding Network

As traffic elements are annotated on the perspective view, it is hard to harness their positional information in the 3D feature space. However, their semantic meaning imposes a great effect. For instance, a road sign indicating the prohibition of left turn usually corresponds to lanes that lay in the middle of the road. This predefined knowledge is beneficial for locating corresponding lanes. We introduce an embedding network to extract semantic information and transform it into a unified feature space to match with centerlines that $\widetilde{Q}_t^i = \texttt{embedding}^i(Q_t^i)$, where $i$ denotes the $i$-th decoder layer. Note that the queries $\widetilde{Q}_t^i$ remain intact in the SGNN. This is intended since imagining traffic elements from centerlines is relatively challenging. Besides, it would be hard to predict traffic elements' attributes in the image feature space if their features have been transformed into another spatial feature space and further updated through interactions with centerlines.

### 3.3.2 Feature Propagation in GNN

In this part, we introduce how topological relationships are modeled and how knowledge from different queries is exchanged. Using GNN, relations can be conveniently formulated as edges in a graph where entities are seen as vertices. However, it is nontrivial in driving scenarios, as there is no explicit constraint or prior knowledge of topology structure. A possible way is to construct a fully connected graph $(V, E)$, with $V = V_l \cup V_t$ and $E \subseteq V \times V$. However, this inevitably increases computational cost and introduces unnecessary information transmission, such as between two traffic elements that are placed subjectively by humans. Instead, we use $G_{ll} = (V_l, V_l \times V_l)$ for lane graph estimation and $G_{lt} = (V_l \cup V_t, V_l \times V_t)$ representing the predicted TE to LC assignments, to guide the information transmission.

In graph $G_{ll}$ and $G_{lt}$, lane queries $Q_l$ are refined by the connected neighbors and corresponding traffic elements. Due to the fact that $Q_l$ and $Q_t$ represent different kinds of objects, the semantic gap still exists. We introduce an adapter layer to combine this heterogeneous information into the information gain denoted as $R$. The overall process in an SGNN layer is formulated as:

$$
\begin{aligned}
Q_l^{i'} &= \texttt{SGNN}_{ll}^i\big(Q_l^i, G_{ll}^{i-1}\big), \\
Q_l^{i''} &= \texttt{SGNN}_{lt}^i\big(Q_l^i, \widetilde{Q}_t^i, G_{lt}^{i-1}\big), \\
R^i &= \texttt{downsample}^i\Big(\texttt{ReLU}\big(\texttt{concat}(Q_l^{i'}, Q_l^{i''})\big)\Big), \\
\widetilde{Q}_l^i &= Q_l^i + R^i.
\end{aligned}
\tag{1}
$$

### 3.3.3 Vanilla Scene Graph

Given the directed lane graph $G_{ll}^{i-1}$ predicted by the previous layer, our goal is to construct a weight matrix $T_{ll}^i$ that controls the flow of messages in the current layer. In this directed graph, messages typically propagate in a single direction, such as from a centerline to its successor. However, the spatial position of a lane can serve as a good indication of the locations of neighboring lanes, which suggests that a bidirectional information exchange could be advantageous. To facilitate this, we augment the weighted adjacency matrix $A_{ll}^{i-1}$ of $G_{ll}^{i-1}$ by incorporating backward edges to construct $T_{ll}^i$, thereby enabling message exchange between two connected centerlines. The process can be formulated as:

$$
T_{ll}^i = \beta_{ll} \cdot \big(A_{ll}^{i-1} + \texttt{transpose}(A_{ll}^{i-1})\big) + I,
\tag{2}
$$

where $T_{ll}^0 = I$ and $I$ denotes the identical mapping for self-loop, $\beta_{ll}$ is a hyperparameter to control the ratio of features propagated between nodes.

In the bipartite graph $G_{lt}$, where only the correspondence between lanes and traffic elements is presented, we utilize features of traffic elements to refine centerline embeddings as follows:

$$T_{lt}^i = \beta_{lt} \cdot A_{lt}^{i-1}, \tag{3}$$

where $T_{lt}^0 = O$ is a matrix in which all entries are zero.

After obtaining the weight matrices, SGNN utilizes the graph convolutional layer (GCN) (Kipf & Welling, 2017) to perform feature propagation among queries:

$$\begin{aligned} Q_l^{i'} &= \text{GCN}_{ll}^i(Q_l^i, T_{ll}^i), \\ Q_l^{i''} &= \text{GCN}_{lt}^i(Q_l^i, \widetilde{Q}_t^i, T_{lt}^i). \end{aligned} \tag{4}$$

### 3.3.4 Scene Knowledge Graph

Though GCN enables feature propagation in the built graphs and treats nodes differently based on their connectivity, the semantic meaning of vertices remains unused. For example, a traffic element indicating to go straight is not equally important as that indicating a red light. To incorporate categorical prior, we design the scene knowledge graph to treat vertices in different classes differently. Fig. 3 illustrates an example process of updating a centerline query $LC_1$ on the given knowledge graph.

On the graph $G_{lt}$, we use $\mathbf{W}_{lt}^i \in \mathbb{R}^{|C_t| \times F_l \times F_t}$ to denote the learnable weights, where $C_t$ describes the attribute set of traffic elements, $F_l$ and $F_t$ are the number of feature channel of LC and TE queries respectively. A centerline query with index $x$ aggregates information from its corresponding traffic elements based on their classification scores:

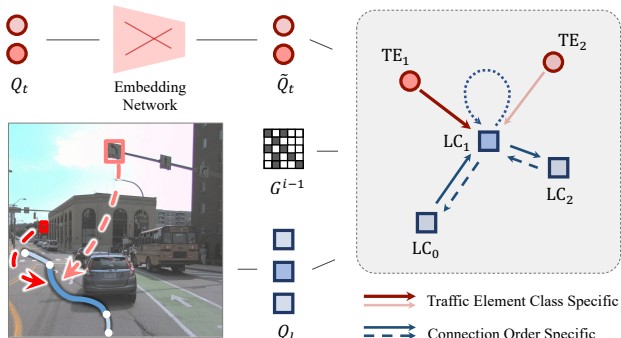

Figure 3: **Scene knowledge graph** illustration. For the centerline colored blue in the left case, related weight matrices in the graph are categorically independent. Different traffic elements and lane-directed connections bring different information to the centerline, which is encoded as a scene knowledge graph on the right.

$$\begin{aligned} K_{lt}^i &= A_{lt}^{i-1}, \\ Q_{l_{(x)}}^{i''} &= \sum_{\forall y \in N(x)} \sum_{\forall c_t \in C_t} \beta_{lt} \cdot S_{t_{(c_t,y)}}^i K_{lt_{(x,y)}}^i \mathbf{W}_{lt_{(c_t)}}^i \widetilde{Q}_{t_{(y)}}^i, \end{aligned} \tag{5}$$

where $N(x)$ outputs the indices of all neighbors of the vertex with index $x$, and $S_t^i \in \mathbb{R}^{|C_t| \times |Q_t^i|}$ represents the classification scores of traffic element queries.

Although all centerlines fall into the same category, the directed connection nature, namely predecessor and successor, still poses an impact on the process of feature propagation. To this end, we formulate the learnable weight matrix for the lane graph as $\mathbf{W}_{ll}^i \in \mathbb{R}^{|C_l| \times F_l \times F_l}$, where $C_l = \{\text{successor}, \text{predecessor}, \text{self-loop}\}$. The centerline queries are further updated by:

$$\begin{aligned} K_{ll}^i &= \text{stack}(A_{ll}^{i-1}, \text{transpose}(A_{ll}^{i-1}), I), \\ Q_{l_{(x)}}^{i'} &= \sum_{\forall y \in N(x)} \sum_{\forall c_l \in C_l} \beta_{ll} \cdot K_{ll_{(c_l,x,y)}}^i \mathbf{W}_{ll_{(c_l)}}^i Q_{l_{(y)}}^i. \end{aligned} \tag{6}$$

### 3.4 Learning

We employ multiple losses to train TopoNet in an end-to-end manner. As depicted in Fig. 2, all heads utilize queries to generate perception and reasoning results. Nevertheless, they are not entirely independent, as the

topology head requires matching results from perception heads. Similar to Transfomer-based networks (Carion et al., 2020; Zhu et al., 2020), the supervision is applied on each decoder layer to optimize the query feature iteratively. The overall loss of the proposed model is $\mathcal{L} = \mathcal{L}_{det_{\mathrm{TE}}} + \mathcal{L}_{det_{\mathrm{LC}}} + \mathcal{L}_{top}$.

**Perception.** Following the head design in DETR (Carion et al., 2020), the TE head predicts 2D bounding boxes with classification scores. Note that for predicting traffic elements, we take $Q_t$ instead of $\widetilde{Q}_t$ to preserve their positional information in the perspective view. The LC head produces 11 ordered 3D points and a confidence score from each centerline query $\tilde{q}_l \in \widetilde{Q}_l$. The ground truth coordinate of centerlines is normalized based on the predefined BEV range. The Hungarian algorithm is utilized to generate matchings between ground truth and predictions for both heads, with the matching cost the same as the loss function. Then task-specific losses $\mathcal{L}_{det_{\mathrm{TE}}}$ and $\mathcal{L}_{det_{\mathrm{LC}}}$ are applied accordingly. Specifically, for the TE head, we employ the Focal loss (Lin et al., 2017b) for classification, an L1 regression loss, and an IOU loss for localization. For centerlines, we use Focal loss and L1 loss as the classification and regression loss, respectively.

**Reasoning.** The topology head reasons pairwise relationships on the given embeddings to predict $A_{ll}$ and $A_{lt}$. Similar to STSU (Can et al., 2021), for a pair of instances, we use two MLP layers to reduce the feature dimension for each instance. Then the concatenated feature is sent into another MLP with a sigmoid activation to predict their relationship. Based on the matching results from perception heads, the ground truth of each pair of embeddings is assigned. In the LC head, we adopt embeddings generated in the SGNN module, i.e., the refined queries $\widetilde{Q}_l$ for lanes and the semantic embeddings $\widetilde{Q}_t$ for traffic elements. Due to the sparsity of the graph, Focal loss is deployed in $\mathcal{L}_{top}$ to deal with the imbalance in sample distribution.

## 4 Experiments

### 4.1 Implementation Details

**Feature Encoding.** We adopt a ResNet-50 (He et al., 2016), which is pre-trained on ImageNet (Deng et al., 2009), with an FPN (Lin et al., 2017a) to obtain multi-scale image features. Following previous works (Zhu et al., 2020; Li et al., 2022b), the output features are from stage $S_{8\times}$, $S_{16\times}$ and $S_{32\times}$ of ResNet-50, where the subscripts $n\times$ indicates the downsampling factor. In the FPN module, the features are transformed into a four-level output with an additional $S_{64\times}$ level. The number of output channels of each level is set to 256. Then we adopt a simplified view transformer with 3 encoder layers proposed in BEVFormer (Li et al., 2022b). Note that we do not use temporal information, and thus the temporal self-attention layer in the BEVFormer encoder is replaced by a deformable attention (Zhu et al., 2020) layer. The size of BEV grids is set to $200 \times 100$, with four different height levels of $\{-1.5m, -0.5m, +0.5m, +1.5m\}$ relative to the ground.

**Deformable Decoder.** For the decoder, we utilize the decoder layer in Deformable DETR (Zhu et al., 2020) that each decoder layer contains three layers: a self-attention layer with 8 attention heads, a deformable attention layer with 8 attention heads and 4 offset points, and a two-layer feed-forward network with 512 channels in the middle. After each operation, a dropout layer with a ratio of 0.1 and a layer normalization is applied. The dimension of initial queries $q = [q_p, q_o] \in Q$ is set to 256, where $q_p$ is utilized to generate the initial reference point, and $q_o$ is the initial object query. The query number for centerlines and traffic elements is set to 200 and 100. The reference points will remain unchanged across different layers.

**Scene Graph Neural Network.** We utilize a simplified version of Graph Convolutional Network (GCN) (Kipf & Welling, 2017) as our GNN layer. Given an input matrix $Q^i \in \mathbb{R}^{N \times C}$, with $N$ representing the number of nodes and $C$ denoting the number of channels, the output of the operation is:

$$Q^{i'} = \sigma\Big(T^i Q^i \mathbf{W}^i\Big), \tag{7}$$

where $\mathbf{W}^i \in \mathbb{R}^{C \times C}$ is the learnable weight matrix, $T^i \in \mathbb{R}^{N' \times N}$ describes the adjacency matrix with $N'$ output nodes, and $\sigma(\cdot)$ is the activation function. Note that the matrix $T$ is inferred without gradients during training. For the traffic element branch, an embedding network is employed before each GNN layer. The embedding network is a two-layer MLP, in which the output channels are 512 and 256. In between the MLP, a ReLU activation function and a dropout layer are included. $\beta_{ll}$ and $\beta_{lt}$ are set to 0.6.

**Prediction Heads.** The prediction head for perception comprises a classification head and a regression head. For the traffic element branch, the classification head is a single-layer MLP, which outputs the sigmoid probability of each class. The regression head is a three-layer MLP with ReLU, which predicts the normalized coordinates of 2D bounding boxes in the form of $\{cx, cy, width, height\}$. For centerline, the classification head consists of a three-layer MLP with LayerNorm and ReLU in between, which predicts the confidence score. The regression head is a three-layer MLP with ReLU, which predicts the normalized point set of $11 \times 3$ for a centerline. To predict topology relationships, relationship heads are applied. Given the instance queries $\widetilde{Q}_a$ and $\widetilde{Q}_b$ with 256 feature channels, the topology head first applies a three-layer MLP:

$$\widetilde{Q}'_a = \texttt{MLP}_\texttt{a}(\widetilde{Q}_a), \ \widetilde{Q}'_b = \texttt{MLP}_\texttt{b}(\widetilde{Q}_b), \tag{8}$$

where the number of output channels is 128. For each pair of queries $\tilde{q}'_a \in \widetilde{Q}'_a$ and $\tilde{q}'_b \in \widetilde{Q}'_b$, the output is the confidence of the relationship, with independent `MLPs` for different types of relationships:

$$conf. = \texttt{sigmoid}\Big(\texttt{MLP}_\texttt{top}\big(\texttt{concat}(\tilde{q}'_a, \tilde{q}'_b)\big)\Big). \tag{9}$$

**Loss.** $\mathcal{L}_{det_{\text{TE}}}$ includes a classification, a regression, and an IoU loss that $\mathcal{L}_{det_{\text{TE}}} = \lambda_{cls} \cdot \mathcal{L}_{cls} + \lambda_{reg} \cdot \mathcal{L}_{reg} + \lambda_{iou} \cdot \mathcal{L}_{iou}$. $\lambda_{cls}$, $\lambda_{reg}$, and $\lambda_{iou}$ are set to 1.0, 2.5, and 1.0, respectively. The classification loss $\mathcal{L}_{cls}$ is a Focal loss. Note that the regression loss $\mathcal{L}_{reg}$ is an L1 Loss calculated on a normalized format of $\{cx, cy, width, height\}$, while the IoU loss $\mathcal{L}_{iou}$ is a GIoU loss computed on the denormalized coordinates. For centerline detection, $\mathcal{L}_{det_{\text{LC}}}$ comprises a classification and a regression loss that $\mathcal{L}_{det_{\text{LC}}} = \lambda_{cls} \cdot \mathcal{L}_{cls} + \lambda_{reg} \cdot \mathcal{L}_{reg}$, where $\lambda_{cls}$ and $\lambda_{reg}$ are 1.5 and 0.025 respectively. Note that the regression loss is calculated on the denormalized 3D coordinates. For topology reasoning, we adopt the same Focal loss but different weights on different types of relationships. The loss $\mathcal{L}_{top}$ is defined as $\lambda_{top_{ll}} \cdot \mathcal{L}_{top_{ll}} + \lambda_{top_{lt}} \cdot \mathcal{L}_{top_{lt}}$, where both $\lambda_{top_{ll}}$ and $\lambda_{top_{lt}}$ are 5.0.

**Training.** The resolution of input images is $2048 \times 1550$, except for the front-view image, which is $1550 \times 2048$ and cropped into $1550 \times 1550$. For data augmentation, $\times 0.5$ resizing and color jitter are used. We adopt the AdamW optimizer (Loshchilov & Hutter, 2018) and a cosine annealing schedule with an initial learning rate of $1 \times 10^{-4}$. TopoNet is trained for 24 epochs with a batch size of 8 with 8 Tesla A100 GPUs.

## 4.2 Dataset and Metrics

We conduct experiments on the OpenLane-V2 benchmark (Wang et al., 2023). The dataset contains topological structures in the driving scenes. Ablation studies are conducted on the *subset_A* of OpenLane-V2.

**Dataset.** Built on top of the Argoverse 2 (Wilson et al., 2021) and nuScenes (Caesar et al., 2020) datasets, the OpenLane-V2 benchmark includes images from 2,000 scenes collected worldwide under different environments. The dataset is split into two subsets, namely *subset_A* and *subset_B*. Each subset contains 1,000 scenes with multi-view images and annotations at $2Hz$. All lanes within $[-50m, +50m]$ along the x-axis and $[-25m, +25m]$ along the y-axis are annotated in the 3D space. Centerlines are described in the form of lists of points that each list comprises 201 points in 3D space. Points of a centerline are ordered according to the spatial position, and the order of points determines the direction of a centerline. Statistically, about 90% of frames have more than 10 centerlines while about 10% have more than 40. Traffic elements follow the typical labeling style in 2D detection that objects are labeled as 2D bounding boxes on the front-view images. Each element is annotated as a 2D bounding box on the front view image, with its attribute. There are 13 types of attributes, including *unknown, red, green, yellow, go_straight, turn_left, turn_right, no_left_turn, no_right_turn, u_turn, no_u_turn, slight_left,* and *slight_right*. The topology relationships are provided in the form of adjacency matrices based on the ordering of centerlines and traffic elements. In the adjacency matrices, an entry $(i, j)$ is positive (i.e., 1) if and only if the elements at $i$ and $j$ are connected.

**Perception Metrics.** The DET score is the typical mean average precision (mAP) for measuring instance-level perception performance. The $\text{DET}_l$ score uses Fréchet distance (Eiter & Mannila, 1994) as the similarity measure, which is very sensitive to line direction and local deviation and thus suitable for evaluating directional lane centerlines. The final score is averaged over match thresholds of $\mathbb{T} = \{1.0, 2.0, 3.0\}$:

$$\text{DET}_l = \frac{1}{|\mathbb{T}|} \sum_{t \in \mathbb{T}} AP_t. \tag{10}$$

Table 1: **Comparison with state-of-the-art methods** on the OpenLane-V2 benchmark. TopoNet outperforms all previous works by a wide margin, especially in directed centerline perception and topology reasoning. *: Topology reasoning evaluation is based on matching results on Chamfer distance. The highest score is bolded, while the second one is underlined.

| Data | Method | $\text{DET}_l\uparrow$ | $\text{TOP}_{ll}\uparrow$ | $\text{DET}_t\uparrow$ | $\text{TOP}_{lt}\uparrow$ | $\text{OLS}\uparrow$ |
|---|---|---|---|---|---|---|
| subset_A | STSU (Can et al., 2021) | 12.7 | 0.5 | 43.0 | _15.1_ | 25.4 |
| | VectorMapNet (Liu et al., 2023a) | 11.1 | 0.4 | 41.7 | 5.9 | 20.8 |
| | MapTR (Liao et al., 2023a) | 8.3 | 0.2 | _43.5_ | 5.9 | 20.0 |
| | MapTR* (Liao et al., 2023a) | _17.7_ | _1.1_ | _43.5_ | 10.4 | _26.0_ |
| | **TopoNet** (Ours) | **28.5** | **4.1** | **48.1** | **20.8** | **35.6** |
| subset_B | STSU (Can et al., 2021) | 8.2 | 0.0 | 43.9 | _9.4_ | 21.2 |
| | VectorMapNet (Liu et al., 2023a) | 3.5 | 0.0 | 49.1 | 1.4 | 16.3 |
| | MapTR (Liao et al., 2023a) | 8.3 | 0.1 | _54.0_ | 3.7 | 21.1 |
| | MapTR* (Liao et al., 2023a) | _15.2_ | _0.5_ | _54.0_ | 6.1 | _25.2_ |
| | **TopoNet** (Ours) | **24.3** | **2.5** | **55.0** | **14.2** | **33.2** |

Note that the defined BEV range is relatively large compared to other lane detection benchmarks (Li et al., 2022a; Liu et al., 2023a), and accurate perception of lanes in the distance is hard. As a result, thresholds $\mathbb{T}$ are relaxed based on the distance between the lane and the ego car. For traffic elements, the $\text{DET}_t$ uses IoU as the similarity measure and is averaged over different types of attributes $\mathbb{A}$ of traffic elements:

$$\text{DET}_t = \frac{1}{|\mathbb{A}|} \sum_{a \in \mathbb{A}} AP_a. \tag{11}$$

**Reasoning Metrics.** The TOP score is an mAP metric adapted from the graph domain. Specifically, given a ground truth graph $G = (V, E)$ and a predicted one $\hat{G} = (\hat{V}, \hat{E})$, it builds a set of vertices $\hat{V}'$ by a projection from $\hat{V}$ such that $V = \hat{V}' \subseteq \hat{V}$, where the Fréchet and IoU distances are utilized for similarity measure among lane centerlines and traffic elements respectively. Inside the predicted $\hat{V}'$, two vertices are regarded as connected if the confidence of the edge is greater than 0.5. Then the TOP score is the averaged vertice mAP between $(V, E)$ and $(\hat{V}', \hat{E}')$ over all vertices:

$$\text{TOP} = \frac{1}{|V|} \sum_{v \in V} \frac{\sum_{\hat{n}' \in \hat{N}'(v)} P(\hat{n}') \mathbb{1}(\hat{n}' \in N(v))}{|N(v)|}, \tag{12}$$

where $N(v)$ denotes the ordered list of neighbors of vertex $v$ ranked by confidence, $\hat{N}'(v)$ denotes the ordered list of predicted neighbors of vertex $v$, and $P(v)$ is the precision of the $i$-th vertex $v$ in the ordered list. The $\text{TOP}_{ll}$ is for topology among centerlines on graph $(V_l, E_{ll})$, and the $\text{TOP}_{lt}$ for topology between lane centerlines and traffic elements on graph $(V_l \cup V_t, E_{lt})$.

**Overall Metrics.** The primary task of the dataset is scene structure perception and reasoning. The OpenLane-V2 Score (OLS) summarizes metrics covering different aspects of it:

$$\text{OLS} = \frac{1}{4} \left[ \text{DET}_l + \text{DET}_t + f(\text{TOP}_{ll}) + f(\text{TOP}_{lt}) \right], \tag{13}$$

where $f$ is the square root function.

### 4.3 Main Results

In Table 1, we compare the proposed TopoNet to several state-of-the-art methods, whose implementation details are described in Appendix A. TopoNet outperforms all previous algorithms by a large margin. As the SOTA map learning method MapTR ignores the direction of centerlines with the permutation-equivalent modeling (Liao et al., 2023a), we additionally evaluate MapTR based on Chamfer distance matching. However, its performance on $\text{DET}_l$, as well as topology metrics, significantly degenerates from TopoNet under

Table 2: **Comparison on centerline perception with a unified feature extractor.** "Topology" denotes that the network is trained with topology supervision.

| Method | Topology | $\text{DET}_l\uparrow$ | $\text{TOP}_{ll}\uparrow$ | $\text{DET}_{l,\text{chamfer}}\uparrow$ | FPS |
|---|---|---|---|---|---|
| STSU (Can et al., 2021) | ✓ | 14.2 | 0.6 | 13.8 | **12.8** |
| VectorMapNet (Liu et al., 2023a) | ✗ | 12.7 | - | 10.3 | 1.0 |
| MapTR (Liao et al., 2023a) | ✗ | 10.0 | - | 21.7 | 11.5 |
| **TopoNet** (Ours) | ✓ | **27.7** | **4.6** | **27.4** | 10.1 |

fair comparison. The large performance gap indicates that reasoning the complex topology raises greater challenges upon merely perceiving presented instances, highlighting the effectiveness of TopoNet's design. All methods achieve similar $\text{DET}_t$, since we adopt the same traffic element detection branch. In more detail, TopoNet possesses slightly superior traffic light detection performance, which indicates that its comprehensive framework is capable of performing heterogeneous feature learning between traffic elements and centerlines, thereby enhancing the performance of $\text{DET}_t$ and $\text{TOP}_{lt}$. On the other hand, since all methods employ a shared backbone, it should be noted that the convergence of traffic light detection could be influenced by other branches, especially when the model struggles to learn centerlines and topological information with a large loss in the LC head. Therefore, given that all methods have the same TE head, our experimental analysis primarily focuses on centerline detection and topology reasoning. Regarding LC-TE topology reasoning, the performance of TopoNet is benefited from its overall superior centerline and traffic element detection performance as well as the proposed SGNN module, in which different entities are modeled differently.

**Comparison on Centerline Perception.** To have a fair comparison, we use a unified backbone and PV-to-BEV transformation module for various SOTA methods on the centerline perception task. We keep the topology supervision for STSU, as it was originally designed for detecting centerlines and their topology relationship. Since VectorMapNet and MapTR initially target laneline detection where there is no relationship between visible lanelines, we alter the supervision from laneline to centerline and ignore topology supervision to preserve their original design choice.

To better align with previous works (Liu et al., 2023a; Liao et al., 2023a), we also provide $\text{DET}_{l,\text{chamfer}}$ with the Chamfer distance as the similarity measure. It does not take the lane direction into account and is thresholded on {0.5, 1.0, 1.5}. As shown in Table 2, TopoNet outperforms other methods on all metrics. We also find that the original design of online mapping approaches struggles with managing lane topology and traffic elements. As shown in Table 1 and Table 2, when the effect from lane topology and traffic elements is removed, MapTR's performance in centerline detection improves from 17.7 to 21.7 on $\text{DET}_{l,\text{chamfer}}$ score. In contrast, TopoNet's performance in centerline detection decreased by 0.8 points on $\text{DET}_l$ due to the removal of the traffic element branch and the lane-traffic element feature interaction in SGNN. This suggests that TopoNet benefits from detecting traffic elements and reasoning the LT topology, attributable to the effective design of our pipeline. Besides, the FPS of TopoNet is 10.1 on an A100 bare machine. Compared to other methods on the same machine with an aligned input size of $512 \times 676$, our method has comparable online efficiency but stronger performance.

**Comparison on BEV Segmentation.** $\text{DET}_l$ is defined to evaluate the validity of each point on a single centerline, ensuring a consistent instance representation of lanes. In contrast, the Intersection over Union (IoU) focuses on pixel-level accuracy in segmentation formulation. It provides a fair comparison of the overall geometric accuracy across various methods with different lane formulations, such as HDMapNet (Li et al., 2022a) and LaneGAP (Liao et al., 2024). Except for HDMapNet, the vectorized centerline predictions of each method are rendered to BEV with a fixed line width of $0.75m$ aligned with the setting in HDMapNet. As shown in Table 3, TopoNet surpasses other methods in terms of IoU. We also conduct a fair comparison with a concurrent work LaneGAP (Liao et al., 2024), which utilizes path-wise modeling

Table 3: **Comparison on BEV segmentation.** When rendering centerlines on the BEV grids, TopoNet also outperforms the previous approach.

| Method | mIoU$\uparrow$ |
|---|---|
| HDMapNet (Li et al., 2022a) | 18.3 |
| STSU (Can et al., 2021) | 24.6 |
| VectorMapNet (Liu et al., 2023a) | 18.9 |
| MapTR (Liao et al., 2023a) | 32.1 |
| LaneGAP (Liao et al., 2024) | 35.0 |
| **TopoNet** (Ours) | **35.1** |

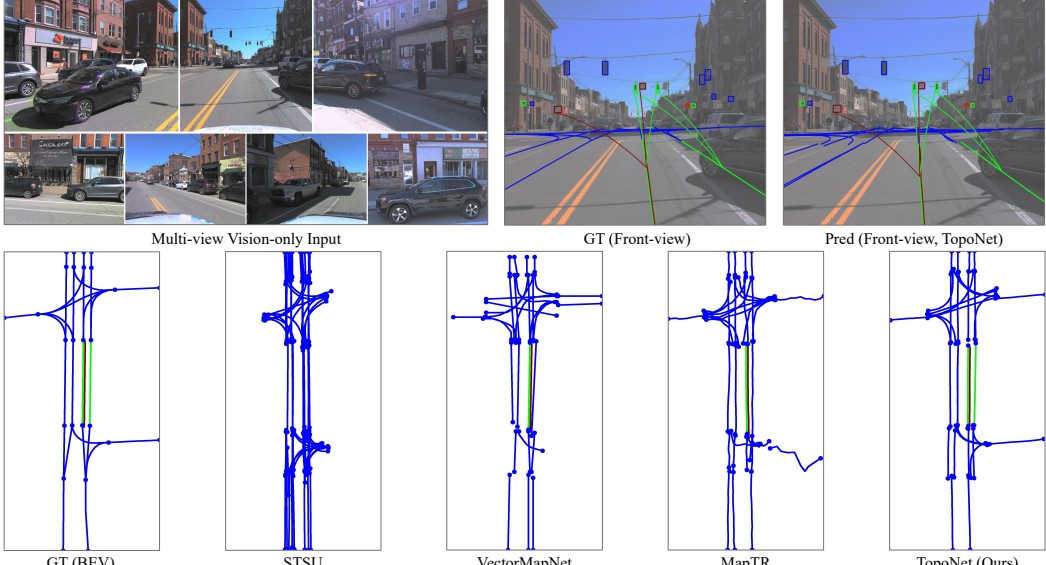

Figure 4: **Qualitative results** of TopoNet and other algorithms on *subset_A* of the OpenLane-V2 dataset. TopoNet achieves superior lane graph prediction performance compared to other SOTA methods in the complex intersection scenario. It also successfully builds all connections between traffic elements and lanes (top right, and correspondingly colored lines in BEV). Colors denote categories of traffic elements.

Table 4: **Ablation on the design of scene graph neural network.** "SG" represents the vanilla scene graph, and "SKG" is the enhanced SGNN with the proposed scene knowledge graph.

| Method | DET$_l$↑ | TOP$_{ll}$↑ | DET$_t$↑ | TOP$_{lt}$↑ | OLS↑ |
|---|---|---|---|---|---|
| Baseline | 25.7 | 4.0 | 47.2 | 20.6 | 34.6 |
| + SG | 27.7 | 3.7 | 48.0 | 20.1 | 35.0 |
| + SKG | **28.5** | **4.1** | **48.1** | **20.8** | **35.6** |

Table 5: **Ablation on feature propagation** in the SGNN. "LL only" denotes aggregation of spatial information from lane connectivity, and "LT only" includes lane-traffic element relationship.

| Method | DET$_l$↑ | TOP$_{ll}$↑ | DET$_t$↑ | TOP$_{lt}$↑ | OLS↑ |
|---|---|---|---|---|---|
| LL only | 27.9 | 3.8 | 47.8 | 20.3 | 35.1 |
| LT only | 27.8 | 3.9 | 47.5 | 20.5 | 35.1 |
| TopoNet | **28.5** | **4.1** | **48.1** | **20.8** | **35.6** |

to represent lane graph. Transforming lane paths into lane pieces in the LaneGAP's post-processing stage necessitates high geometric accuracy, making it unsuitable for evaluation using DET$_l$. This method achieves similar performance to TopoNet in terms of IoU. However, we note that piece-wise modeling of TopoNet can effectively capture the precise locations of lane splits or merges, as well as the topology between lanes and traffic elements, making it more suitable for practical applications.

### 4.4 Ablation Study

**Effect of Design in Scene Graph Neural Network.** We alternate the proposed network into a baseline without feature propagation by downgrading the SGNN module to an MLP and supervising topology reasoning at the final decoder layer only. The concatenation and down-sampling operations, as well as the traffic element embedding, are also removed. As illustrated in Table 4, the proposed SKG outperforms models in other settings, demonstrating its effectiveness for topology understanding. Compared to the SG version, the scene knowledge graph provides an additional improvement of 0.8% for centerline perception, owning to the predefined semantic prior encoded in the categories of traffic elements. The improvement of traffic element detection and topology reasoning is also consistent. Given that transformers are widely regarded as a variant of GNN, this also reveals that explicitly designing the feature interaction between queries within a transformer decoder can further enhance performance, especially when instances have a strong correlation.

**Effect on Feature Propagation.** In the "LL only" setting, we set the $\beta_{lt}$ parameter to 0. Similar to the baseline, we remove the concatenation and down-sampling operations, as well as the traffic element

Table 6: **Ablation on the number of GNN layers** in the scene knowledge graph. Model performance drops as the number of SGNN layers increases.

| # GNN | $\text{DET}_l\uparrow$ | $\text{TOP}_{ll}\uparrow$ | $\text{DET}_t\uparrow$ | $\text{TOP}_{lt}\uparrow$ | OLS↑ |
|---|---|---|---|---|---|
| 1 | **28.5** | **4.1** | **48.1** | **20.8** | **35.6** |
| 2 | 27.9 | 4.0 | 47.5 | 20.9 | 35.3 |
| 3 | 20.4 | 0.5 | 46.1 | 15.7 | 28.3 |

Table 7: **Ablation on edge weight** in the scene knowledge graph. The magnitude of edge weight has an impact on model performance.

| Weight | $\text{DET}_l\uparrow$ | $\text{TOP}_{ll}\uparrow$ | $\text{DET}_t\uparrow$ | $\text{TOP}_{lt}\uparrow$ | OLS↑ |
|---|---|---|---|---|---|
| 0.5 | 28.4 | 4.0 | 47.7 | **20.8** | 35.4 |
| 0.6 | **28.5** | **4.1** | **48.1** | **20.8** | **35.6** |
| 0.7 | 27.3 | **4.1** | 47.7 | 20.7 | 35.1 |

embedding. For "LT only", we set the $\beta_{ll}$ parameter to 0, while other modules remain intact. Results are reported in Table 5. In the "LL only" setting, the drop on $\text{TOP}_{lt}$ demonstrates the importance of the graph $G_{lt}$. Besides, it can be observed that the performance of $\text{DET}_l$ experiences a certain decline under this setting as well. This might result from the lack of traffic element features' guidance for lane centerline detection within intersections. Compared to non-intersection areas, there is a higher number of centerlines within intersections, while they lack distinct lane marking features and require traffic elements' guidance.

With the "LT only" design, $\text{DET}_l$ degenerates when removing the graph $G_{ll}$, showing the importance of feature propagation between centerline queries. These experiments show that both branches are necessary for achieving satisfactory model performance on the primary task.

**Effect on the Number of GNN Layers.** Though GNN is beneficial for propagating features in the knowledge graph, raising the number of GNN layers leads to degenerated performance. As shown in Table 6, SGNN with a single GNN layer achieves the best performance. The reason is that a GNN layer increases the similarity of adjacent vertices, causing the over-smoothing effect commonly associated with GNNs.

**Effect on Edge Weight.** Edge weight in the scene knowledge graph represents how much information is propagated through the SGNN layers. We choose the edge weights around the reciprocal of the average number of lane neighbors in the dataset to balance the contributions of features from neighboring nodes and the central node itself. In Table 7, 0.6 corresponds to the most appropriate ratio.

### 4.5 Qualitative Analysis

We provide a qualitative comparison on *validation* set in Fig. 4. We show the raw output of each method, without the post-processing technique suggested in STSU (Can et al., 2021), to avoid the potential introduction of accumulated inaccuracies and misalignment with quantitative evaluation. TopoNet predicts most centerlines correctly and constructs a lane graph in BEV. Yet, prior works fail to output all entities or get confused about their connectivity. More visualizations are provided in Appendix C.

## 5 Conclusion and Future Work

In this paper, we discuss abstracting driving scenes as topology relationships of various entities and propose TopoNet, to address the problem. Importantly, our method models feature interactions via the graph neural network architecture and incorporates traffic knowledge in heterogeneous feature spaces with the knowledge graph-based design. Our experiments on the large-scale OpenLane-V2 benchmark demonstrate that TopoNet excels previous SOTA approaches on perceiving and reasoning about the driving scene topology under complex urban scenarios.

**Limitations and Future Work.** Benefiting from the query-based design for feature propagation, TopoNet performs well in outputting positive predictions. However, post-processes such as merging or pruning are still needed to produce clean output, just as other lane topology works (Can et al., 2021; Büchner et al., 2023). The topic of incorporating the merging ability with auto-regressive or other association mechanisms deserves future exploration. Meanwhile, it will be interesting to see if more categories of traffic elements and a more sophisticated knowledge graph design will make any advances.

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

# Appendix

## A    Re-implementation of SOTA Methods

Since there are no prior methods for the task of driving scene understanding, we adapt three state-of-the-art algorithms which are initially designed for lane graph estimation or map learning: STSU (Can et al., 2021), VectorMapNet (Liu et al., 2023a), and MapTR (Liao et al., 2023a). To ensure a fair comparison, we employ the same input resolution, the same ResNet-50 image backbone, and the same FPN neck for extracting features from surrounding images. Additionally, we incorporate a Deformable DETR head specifically for traffic elements, aligning all settings with TopoNet. As for the topology reasoning module, we treat it differently based on their own modeling concepts of instance query. The topology heads for each method are the same MLPs as in TopoNet. All the methods are trained for 24 epochs to ensure a fair comparison.

**STSU.** The original model predicts centerlines and their relationships under the monocular setting. It employs a BEV positional embedding and a DETR head to predict three Bezier control points for each centerline, and uses object queries in the decoder to predict the connectivity of centerlines. To adapt to the multi-view inputs, we re-implement STSU by computing and concatenating the BEV embedding of images from different views. The concatenated embedding is then fed into the DETR encoder. We retain the original DETR decoder to predict the Bezier control points, which are interpolated into 11 equidistant points as outputs. The lane-lane relationship prediction head of STSU is preserved as well.

**VectorMapNet** utilizes a DETR-like decoder to estimate key points and an auto-regressive module to generate detailed graphical information for a map elements instance, such as lanelines and pedestrian crossings. We supervise VectorMapNet's decoder with centerline labels to adapt to the OpenLane-V2 task. The perception range is defined as $\pm 30m \times \pm 15m$ in the original setting, and we expand it to $\pm 50m \times \pm 25m$. The centerline outputs of VectorMapNet are interpolated to 11 equidistant points during the prediction process. For topology prediction, we use the key point object queries in the VectorMapNet decoder as instance queries of centerlines. We implement the modification on the given codebase of VectorMapNet while retaining other settings. However, due to their lack of support for 3D centerlines, we only predict 2D centerlines in the BEV space and ignore the height dimension during training and evaluation.

**MapTR.** MapTR directly predicts polylines with a fixed number of points using a DETR-like decoder. It utilizes a hierarchical query, representing each line instance with multiple point queries and one instance query. For topology prediction, we use the average of the hierarchical queries of an instance in the MapTR decoder as the instance query of a centerline. The traffic element head and the topology head are with the same setting as in TopoNet. We align the original backbone setting with TopoNet. The perception region is also expanded to $\pm 50m \times \pm 25m$. The implementation is also done on the open-source codebase of MapTR with other settings retained. Due to the lack of support for 3D centerlines, we only predict 2D centerlines in BEV and ignore the height dimension during training and evaluation.

## B    More Experiments

Table 8: **Ablation on traffic element embedding.** TE embedding is necessary to deal with inconsistency in the feature space of different queries.

| Method | $\text{DET}_l\uparrow$ | $\text{TOP}_{ll}\uparrow$ | $\text{DET}_t\uparrow$ | $\text{TOP}_{lt}\uparrow$ | OLS↑ |
|---|---|---|---|---|---|
| w/o embedding | 28.4 | **4.1** | 46.9 | 20.5 | 35.2 |
| TopoNet | **28.5** | **4.1** | **48.1** | **20.8** | **35.6** |

**Effect on Traffic Element Embedding.** In the "w/o embedding" setting, we remove the traffic element embedding network and use $Q_t$ as the input of SGNN directly. As shown in Table 8, removing the embedding results in a 1.2% performance drop in traffic element recognition. The reason is that TE queries contain a large amount of spatial information in the PV space due to the 2D detection supervision signals,

Table 9: **Comparison with the winning methods in the CVPR 2023 Autonomous Driving Challenge**. The upper part is the Leaderboard on the OpenLane-V2 *test* split. The down part is the performance on the *val* split with ResNet-50 backbones. The listed teams utilized non-shared backbones for the lane and traffic element branches. "# Params." refers to the total number of backbone parameters. "*": using post-processing on the topology prediction. TopoNet surpasses the third-place method on the overall performance, with only 25M backbone parameters and 24 epoch training.

| Data | Team & Method | Backbone | # Params. | Epoch | $DET_l\uparrow$ | $TOP_{ll}\uparrow$ | $DET_t\uparrow$ | $TOP_{lt}\uparrow$ | OLS↑ |
|---|---|---|---|---|---|---|---|---|---|
| *test* | MFV (Wu et al., 2023) (1st) | ViT-L + CSPNet (YOLOv8x) | 375M | 48 + 20 | 35.8 | 22.5* | 79.7 | 33.5* | 55.2 |
| | Victory (Lu et al., 2023) (2nd) | Swin-S + Swin-S | 100M | Unknown | 21.8 | 13.2* | 72.5 | 22.6 | 44.6 |
| | PlatypusWhispers (Kalfaoglu et al., 2023) (3rd) | RegNetY-800mf + ConvNext-B | 95M | 40 + 30 | 22.1 | 6.0 | 70.6 | 15.7 | 39.2 |
| | **TopoNet (Ours)** | **ResNet-50 (shared)** | **25M** | **24** | 25.8 | 10.1* | 59.5 | 23.7* | 41.4 |
| *val* | MFV (Wu et al., 2023) (1st) | ResNet-50 (LC only) | 25M | 20 | 18.2 | - | - | - | - |
| | PlatypusWhispers (Kalfaoglu et al., 2023) (3rd) | ResNet-50 + ResNet-50 | 50M | 24 + 24 | 22.1 | 5.8 | 58.2 | 15.5 | 36.0 |
| | **TopoNet (Ours)** | **ResNet-50 (shared)** | **25M** | **24** | **28.5** | **4.1** | 48.1 | **20.8** | 35.6 |

resulting in significant inconsistencies in the feature spaces. In all, the experiments demonstrate that TE embedding effectively filters out irrelevant spatial information and extracts high-level semantic knowledge to help centerline detection and lane topology reasoning.

**Comparison on the OpenLane-V2 leaderboard.** We compare TopoNet with the winning methods in the CVPR 2023 Autonomous Driving Challenge in Table 9. The leading methods of the competition employ various tricks to maximize the performance, such as stronger and non-shared backbones, longer training epochs, training on the validation set, extensive hyper-parameter tuning, and complex data augmentation and post-processing strategies. Because most methods in the competition employ SOTA 2D detection approaches and non-shared backbones, we primarily compare the effectiveness of TopoNet in the context of lane graph perception. After utilizing the post-processing technique of MFV (Wu et al., 2023) on lane-lane topology prediction, TopoNet achieves a $DET_l$ score of 25.8 and a $TOP_{ll}$ of 10.1 on the OpenLane-V2 *test* set, achieving superior centerline detection performance compared to the second-place method. TopoNet employs a shared ResNet-50 backbone, being up to $15\times$ smaller in backbone parameter size than the awarded methods, demonstrating great training efficiency.

We further provide the comparison on the *validation* split, where these methods report performance with a ResNet-50 backbone and without most tricks. MFV, the first-place method in the competition, achieves a $DET_l$ score of 18.2, and the third-place team PlatypusWhisperers (Kalfaoglu et al., 2023) gets a $DET_l$ score of 22.1. With less data augmentation and hyper-parameter tuning, TopoNet achieves a much higher $DET_l$ score of 28.5, surpassing all methods above. These fair comparisons on the validation set well demonstrate the effectiveness of TopoNet's pipeline.

# C   More Visualization

We provide additional qualitative comparisons on *subset_B* of OpenLane-V2 in Fig. 5. Fig. 6 shows a case where a bus occludes the intersection in the front view image. TopoNet fails to predict lanes and the topology, especially those in the left half of the crossing. A large-scale dataset and learning techniques, such as active learning, would solve such failure cases in a real-world deployment.

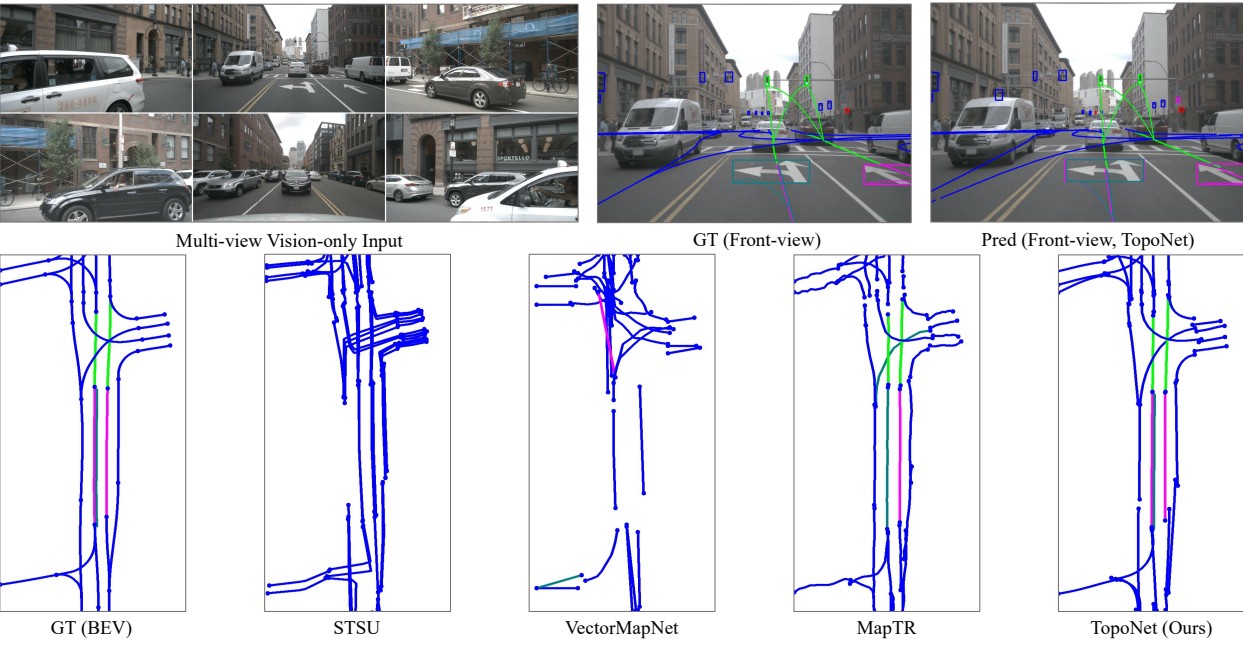

Figure 5: **Qualitative results** of TopoNet and other algorithms on *subset_B* of the OpenLane-V2 dataset. Colors denote categories of traffic elements.

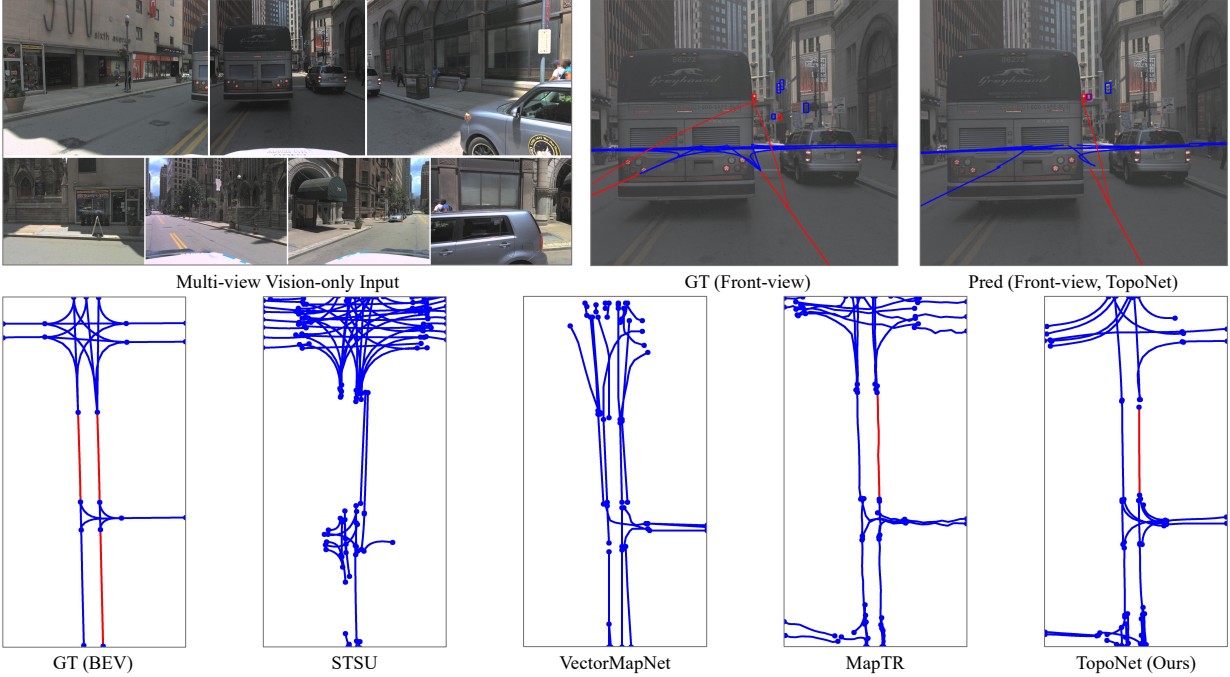

Figure 6: **Failure case under large-area occlusion.** TopoNet fails to predict centerlines and the lane graph in the intersection with a large bus colluding in front. Note that the relationship between the left lane and the red light is an incorrect annotation where our algorithm reasons about the direction of the left lane and avoids the false positive prediction.

