# OpenReview forum: "Graph-based Topology Reasoning for Driving Scenes"
_TMLR — Rejected by TMLR_

### Review · Reviewer_YasG · 2024-08-14

**Summary Of Contributions:**

The paper proposes a new architecture for constructing road graphs from perceptual inputs in driving contexts. The architecture is primarily based on GNNs and contains several novel mechanisms designed to help with this specific tasks. The authors claim substantial improvements of state of the art methods on a recent public benchmark OpenLane-V2.

**Audience:**

Yes

**Claims And Evidence:**

Yes

**Requested Changes:**

In my opinion, the main issue is that the proposed architecture is somewhat complicated and it's not easy to check whether the text contains enough detail to reproduce the results. The authors promise the code will be released, but since we can't see it, there's no way to tell whether the code would be sufficient to reproduce the results. If the authors release the code and it looks complete and readable, I would be comfortable recommending acceptance. Without it, I'm not really sure that the paper is sufficiently interesting to the community.

I would also appreciate some clarification regarding the current state of the art results on the OpenLane-V2 benchmark. I can't reconcile the leaderboard with the tables in the paper. I was looking at the leaderboard linked by the GitHub repo.
https://github.com/OpenDriveLab/OpenLane-V2

The manuscript is overall easy to follow, but it contains a fair number of odd phrases that I can't make sense of. Examples from the first page include "road genome", "reasoning method is vacant", "tabula-rasa resolution", and "in the academy".

**Strengths And Weaknesses:**

The problem considered is very important for autonomous driving and largely underexplored to date. The paper is clear and easy to follow. The contribution is very specificly improving the network architecture used for this particular task, and the paper is accordingly light on theory and heavy on evaluation, including qualitative, quantitative, and ablations. It looks like a solid, if somewhat incremental, contribution to the field.

---

> ### Author Response · Authors · 2024-08-28
> **Response from Authors**
>
> Thank you for your valuable and positive feedback! We answer all the questions and concerns raised in your review hereafter.
>
> ---
>
> > *${\color{BrickRed}Question 1:}$* The main issue is that the proposed architecture is somewhat complicated and it's not easy to check whether the text contains enough detail to reproduce the results. The authors promise the code will be released, but since we can't see it, there's no way to tell whether the code would be sufficient to reproduce the results.
>
> Thanks. We kindly understand your concern and have provided implementation details in Section 4.1, including model structures like FPN output features, downsample ratio, and decoder head, and hyperparameters like the number of queries' dimensions, number of queries, and edge weights in GNN layers, etc.
>
> Additionally, we prepare and provide our code in an anonymous repository for the review stage: https://github.com/anonymouslab-ai/TopoNet-anonymous. We will publicly share it with the community.
>
> ---
>
> > *${\color{BrickRed}Question 2:}$* Some clarification regarding the current state of the art results on the OpenLane-V2 benchmark. I can't reconcile the leaderboard with the tables in the paper.
>
> Our task corresponds to the OpenLane Topology Challenge at CVPR 2023. The figure of the leaderboard in the repo was maintained by the challenge organizer, and we refer to the leaderboard on the test server at: https://eval.ai/web/challenges/challenge-page/1925/leaderboard/4549.
>
> In fact, we have provided the results in Appendix Section B (Table 9) in our submitted manuscript. As mentioned in the paragraph below Table 9, compared to the awarded methods in the CVPR 2023 Autonomous Driving Challenge which employed various tricks to maximize the performance, TopoNet achieves a DET$\_{l}$ score of 25.8 and a TOP$\_{ll}$ of 10.1 on the test set. Our method achieves better centerline detection performance compared to the second-place method with a 15x smaller in backbone parameter size. With a fair comparison, TopoNet can get a DET$\_{l}$ score of 28.5 on the validation split, surpassing the awarded methods. Due to the limited space, we put the table in the supplementary material.
>
> ---
>
> > *${\color{BrickRed}Question 3:}$* Odd phrases.
>
> Sorry for the confusion. We have carefully refined the manuscript and asked professional native English speakers to polish it. We reply to the wording questions in your review below.
>
> 1. "Road genome": This expression is inspired by the visual genome [a] mainly, which describes the structured topology in images. It also originates from the large-scale dataset TopoNet is validated on, OpenLane-V2, which was previously named Road Genome (https://arxiv.org/abs/2304.10440v1). We have revised the word as road structure or road topology for a clearer understanding.
>
> 2. "Reasoning method is vacant": Agreed. We have rephrased the expression to "reasoning method is still absent".
>
> 3. "Tabula-rasa resolution": We would like to indicate the resolution mentioned is very straightforward. It does not involve a novelly designed centerline detection method. We have revised the sentence for a better understanding.
>
> 4. "In the academy": Thanks for finding the typo. We have revised it as "in the academia".
>
> [a] Krishna, R., Zhu, Y., Groth, O. et al. Visual Genome: Connecting Language and Vision Using Crowdsourced Dense Image Annotations. Int J Comput Vis 123, 32–73 (2017). https://doi.org/10.1007/s11263-016-0981-7

---

### Review · Reviewer_Eqrf · 2024-08-18

**Summary Of Contributions:**

TopoNet tries to address the crucial problem of topology reasoning in driving scenes with an end-to-end manner. The paper tries to address the problem where both the centerlines and traffic elements need to be detected and extracted in an instance level, and also that the topological relationships between them should be extracted. The positional information is important for centraline but less for traffic elements while the semantic meaning for traffic elements is more important. The network takes as input multi-view images and process them through four stages of encoding and decoding to output the traffic elements and central lines information, as well as their topological relationships. TopoNet has two important sub-networks, the Scene graph neural network (SGNN) and the Scene Knowledge Graph (SKG). TopoNet encodes multi-view images into multi-scale image features, and converts them into instance-level embeddings. It then uses SGNN to refine centerline queries and topological aspects. Decoders and SGNN layers stacked iteratively to obtain local and global features in a sequential fashion. Task specific heads takes these refined queries to output prediction results. SKG mitigates the issue where semantic meanings of vertices are less explored. SKG treats vertices in different classes differently. The learning is based on combined loss function consisting of detection of traffic elements loss, centerline detection loss and topological loss. The results indicate that the proposed topoNet exceeds baselines on the OpenLane-v2 benchmark by a wide margin.

**Audience:**

Yes

**Broader Impact Concerns:**

This approach could have significant impact on the society as self-driving are more adopted world-wide. Definitely, acquiring roadgraphs from vision only input can be very useful, which reduces the sensor requirement and makes self-driving more accessible and economically efficient.

**Claims And Evidence:**

Yes

**Requested Changes:**

Provides rationale on why the ablation studies choose the ablation parameters, including the number of GNN layers, why it is 1,2,or 3. Also explain why the edge weights are from 0.5 , 0.6 to 0.7.

Provides explanation on insights whether the differences of the results between MapTR and the proposed TopoNet is that significant. Do they affect driving performance?

**Strengths And Weaknesses:**

Strength of the work: it proposes to use transformers and SGNN, SKG to solve the problem of detecting traffic elements and topological relationships. The approach achieves significant improvements over existing baselines. It also conducted ablation study to show the effect of design in SGNN such as the number of GNN layers and edge weight in SKG.

Weakness: the metrics defined in the paper may not optimally reflect the importance of lane detection and traffic elements detection. For example, in the visualized qualitative results in figure 4, the results of MapTR look very similar to TopoNet, only some minor differences. Though on the number the MapTR method is worse. Are these differences significant for driving? Also the ablation studies pick up some ablation options such as different number of GNN layers but does not explain why these numbers are picked. For example, why the edge weight is from 0.5, 0.6 to 0.7?

---

> ### Author Response · Authors · 2024-08-28
> **Response from Authors**
>
> Thank you for your helpful review! We address your questions below.
>
> ---
>
> > *${\color{BrickRed}Question 1:}$* The metrics defined in the paper may not optimally reflect the importance of lane detection and traffic elements detection. Insights whether the differences of the results between MapTR and the proposed TopoNet is that significant. Do they affect driving performance?
>
> Thanks for the insightful question. Note that the qualitative results in Figure 4 do not show the direction of centerlines for clear visualization and simplicity. The performance regarding centerlines' directions can be reflected in metrics of DET$\_{l}$.
>
> - **The reason why MapTR performs worse on DET$_l$.** For centerline detection, we use DET$\_l$ from the OpenLane-V2 benchmark as the main metric. DET$\_l$ uses Frechet distance for mAP matching, which addresses the importance of centerline direction. As analyzed in Section 4.3, MapTR is designed to detect **undirected** lines, resulting in low DET$\_l$ performance on the OpenLane-V2 benchmark.
>
> - **Additional comparisons on DET$_{l,\text{chamfer}}$ and mIoU.** To fairly compare with MapTR, we conduct comparison on DET$\_{l,\text{chamfer}}$ with Chamfer distance for mAP matching in Table 2. MapTR reaches a DET$\_{l,\text{chamfer}}$ score of 21.7. In this setting, TopoNet achieves 27.4 which is also better than other methods. We also conduct a comparison on mIoU in Table 3, which is a closer representation of the quantitive results. In this comparison, MapTR reaches 32.1 and TopoNet reaches 35.1.
>
> - **Why using DET$\_l$ with Frechet distance is important.** Precise centerline detection results are crucial for various downstream tasks, including motion prediction and planning. The Fréchet emphasizes the direction of centerlines compared to other similarity measurements like Chamfer distance. Any significant deviation in either the horizontal or vertical direction on any point of the line can lead to a mismatch. DET$\_l$ uses Fréchet distance with thresholds 1m, 2m, and 3m for matching. Given that most lanes have a width of approximately 4 meters, a horizontal deviation of 2m can place the centerline outside the lane, which is unacceptable for downstream applications. Utilizing a strict metric like DET$\_l$ allows for an assessment of whether the model's output is reliable for direct use.
>
> **Action items.** We have revised Sections 4.2 and 4.3 to incorporate the above discussions.
>
> ---
>
> > *${\color{BrickRed}Question 2:}$* Rationale on why the ablation studies choose the ablation parameters, including the number of GNN layers, why it is 1,2,or 3. Also explain why the edge weights are from 0.5 , 0.6 to 0.7.
>
> - For **number of GNN Layers**: In our design, GNN layers are integrated after each transformer decoder layer, which is stacked in six layers. Introducing a single GNN layer leads to a performance improvement compared to the baseline without GNN layers. However, as we increase the number of GNN layers to more than one, we observe a decline in performance, likely due to the over-smoothing effect commonly associated with GNNs. Based on these observations, we choose not to add more GNN layers in the ablation.
>
> - For **edge weights**: The edge weights are employed to balance the contributions of features from neighboring nodes and the central node itself. For example, if a lane has two neighbors, the potential strategy to balance feature learning is to use an edge weight of 1/3 for each neighbor and itself. Following this, we initially choose the edge weights around the reciprocal of the average number of lane neighbors and subsequently tune them. In the OpenLane-V2 dataset, the average number of lane neighbors is 1.89, yielding a reciprocal value of approximately 0.53. Thereafter, we ablate the edge weights with the numbers in the table.
>
> **Action items.** We have included the above discussions in the revision to the paragraph "Effect on the Number of GNN Layers" and "Effect on Edge Weight" respectively.

---

### Review · Reviewer_m1ot · 2024-08-19

**Summary Of Contributions:**

The key idea seems to be to take multi-view posed images from cameras on a car, encode these into feature maps in perspective-view and bird's eye view (BEV), then apply a traffic light detector on the perspective-view feature maps (output in the form of box coordinates, presumably at each feature map cell) and a centerline-detector in the BEV feature maps (output in the form of 11 3D points total), then do some graph convolutions between these two outputs, and output refined versions of those estimates. An important novelty here is to create an adjacency matrix that gates the message-passing between these outputs, and this matrix appears to be handcrafted using rules.

**Audience:**

Yes

**Claims And Evidence:**

No

**Requested Changes:**

I think this paper needs very heavy rewriting. I think ChatGPT may actually be useful for this sort of job, but it is really critical for the authors to make sure that the paper text describes what they intend to describe. I am quite sure that the paper idea in the authors' minds is much more intelligible than the PDF included for review here.

**Strengths And Weaknesses:**

This paper does not make sense. I think it may have been written in a different language and then sent through some automatic translator, and the outcome of this process is ultimately only approximately understandable. As written, the paper apparently focuses on understanding "the road genome" (which does not exist), and claims this problem is itself "intelligent" and "intellectual" (which it cannot be), and proceeds to propose an end-to-end framework which finds "traffic knowledge beyond conventional perception tasks" (which I don't know how to interpret), "and achieves times of performance in terms of the challenging topology-reasoning task" (which is not a measurable result). This is only a small sampling of the paper text -- arbitrary sections can be picked apart like this. I can do my best to guess at the intended meaning, but I feel this is not entirely appropriate to do as an objective reviewer.

According to the evaluations, the proposed method outperforms prior work on the same task, and the provided ablations show multiple interesting factors of performance. I would have liked to see an additional ablation for the adjacency matrix, since that part seems central to the approach.

The adjacency matrix description is not super clear because the first mention of it assumes that it was already defined (sec. 3.3.3, "Given the adjacency matrix A_{ll}^{i-1}...").

In this review, where the system asks "Would at least some individuals in TMLR's audience be interested in knowing the findings of this paper", I am clicking "Yes" to optimistically indicate that I believe the science here is probably good and interesting, and if the paper were heavily rewritten so as to accurately reflect the work done, some of the TMLR audience would be interested.

---

> ### Author Response · Authors · 2024-08-28
> **Response from Authors**
>
> Thank you for your comments! We address your concerns below.
>
> ---
>
> > *${\color{BrickRed}Question 1:}$* I think this paper needs very heavy rewriting. I think ChatGPT may actually be useful for this sort of job, but it is really critical for the authors to make sure that the paper text describes what they intend to describe.
>
> Thanks for your comment. We guarantee that the paper is fully written in English from scratch. Sorry for the confusion caused during your reading. We first reply to the specific wording questions in your review below.
>
> 1. **Road genome**: This expression is inspired by the visual genome [a] mainly, which describes the structured topology in images. It also originates from the large-scale dataset TopoNet is validated on, OpenLane-V2, which was previously named Road Genome (https://arxiv.org/abs/2304.10440v1). We have revised the word as road structure or road topology for a clearer understanding.
>
> 2. **Intelligent and intellectual**: We employ these words because that our method reasons the road topology besides the popular detection task. We humbly believe the topology understanding reflects a degree of intelligence. However, we agree with the reviewer that it does not feature intelligence compared to areas like large language models. We have rephrased the words.
>
> 3. **Abstracting traffic knowledge beyond conventional perception tasks**: Here we refer conventional perception tasks to problems like traffic light detection and laneline detection in autonomous driving, which has been explored for a few years. In our work, TopoNet is capable of building the topological structure between lanes and traffic elements, besides detecting them. We have revised them to a more straightforward expression.
>
> 4. **Achieves times of performance in terms of the challenging topology reasoning task**: The performance highlighted here refers to the topology-related metrics, TOP$\_{ll}$ and TOP$\_{lt}$. As shown in the results in Table 1, TopoNet outperforms previous methods by a large margin. Expressively, TOP$\_{ll}$ of TopoNet is 4.1 and MapTR's is 1.1; TOP$\_{lt}$ of TopoNet is 20.8 and STSU's is 15.1. We have revised the expression.
>
> In the meantime, we would also like to note that Reviewer YasG mentions that our work is "clear and easy to follow". We sincerely thank you for your review and will carefully improve our manuscript based on your comments.
>
> **Action items.** We have used tools including ChatGPT to refine the manuscript, revised some words for accurate expressions, and asked professional native English speakers to polish it. Important revisions have been marked in blue in the manuscript.
>
> [a] Visual Genome: Connecting Language and Vision Using Crowdsourced Dense Image Annotations. IJCV 2017.
>
> ---
>
> > *${\color{BrickRed}Question 2:}$* The adjacency matrix description is not super clear because the first mention of it assumes that it was already defined (sec. 3.3.3, "Given the adjacency matrix A_{ll}^{i-1}..."); An additional ablation for the adjacency matrix.
>
> Thanks for the question. First, we would like to clarify that the adjacency matrix $A_{ll}^{i-1}$ described in Section 3.3.3 and 3.3.4 is used to represent the connectivity of $G_{ll}^{i-1}$ in implementation. Expressively, $G_{ll}^{i-1}$ is the directed **graph** predicted from the previous decoder layer, including nodes and edges; while $A_{ll}^{i-1}$ is the **adjacency matrix/tensor** to represent the existence of edges in $G_{ll}^{i-1}$. We explicitly differentiate these two notations due to the broader meaning of a graph $G_{ll}^{i-1}$. Note that this also applies for $A_{lt}^{i-1}$ and $G_{lt}^{i-1}$. We have revised the descriptions in Section 3.1 and Section 3.3 accordingly for a clearer understanding.
>
> **Ablation**
>
> The adjacency matrices $A_{ll}$ and $A_{lt}$ are predicted and then supervised by the ground truth from the dataset. All baseline methods adopt the same supervision. Meanwhile, we agree that it is crucial to our method.
> In both the vanilla scene graph and scene knowledge graph, we use the structure stored in $A_{ll}$ and $A_{lt}$ to control the information transmission in SGNN.
>
> We conduct multiple ablations on how the $A_{ll}$ and $A_{lt}$ are used in SGNN.
> - In Table 4, the matrices in SG and SKG are multiplied without or with a learnable weight matrix storing prior knowledge.
> - In Table 5, the edge weight of $A_{lt}$ and $A_{ll}$ is set to 0 in "LT only" and "LL only", respectively.
> - In Table 7, the choice of edge weight of these matrices is explored.
>
> As requested, we further conduct additional experiments on the direction of information transmission for $A_{ll}$ in the rebuttal. The results show that the design of bi-directional information transmission brings reasonable performance gain, for effective information exchange between lane neighbors.
>
> |Method|DET$\_l$|TOP$\_{ll}$|DET$\_{t}$|TOP$\_{lt}$|OLS|
> |-|-|-|-|-|-|
> |single-directional|28.1|4.0|47.2|20.6|35.2|
> |bi-directional (TopoNet)|28.5|4.1|48.1|20.8|35.6|

---

### Author Response · Authors · 2024-08-28
**General Response from Authors**

We want to express our sincere gratitude for the detailed reviews of our work and the valuable comments and feedback. We have answered each review individually, providing clarifications regarding questions or concerns. Furthermore, we have uploaded the revised version of our paper, in which all the important revisions in the main paper have been marked in blue.

We remain available to answer any remaining questions. Once again, we thank you for your consideration and all the time invested in this review process.

---

### Decision · Action_Editor_JziZ · 2024-11-05

**Recommendation:** Reject

**Comment:**

The reviewers appreciate the importance of estimating the topology of road
networks, particularly in the context of self-driving. Further, they emphasize
that the experimental results demonstrate the performance gains provided by the
proposed framework, while the ablation studies offer interesting insights into
the method. As Reviewer m1ot notes, an additional ablation of the adjacency
matrix would strengthen the evaluation.

A primary concern with the initial submission is that the writing is difficult
to understand. This makes it hard to appreciate the contributions of the
end-to-end framework and to reproduce the results. The authors made a concerted
effort to address these issues during the discussion phase. This included
sharing an anonymized version of the code, which the reviewers and AE
appreciate.

The AE acknowledges some disagreement among the reviewers with regard to the
writing clarity, notably between Reviewers m1ot and YasG. Reviewer YasG states
that the paper is "clear and easy to follow", but then notes that "it contains
a fair number of odd phrases that I can't make sense of". Additionally, the
reviewer suggests that the contributions are incremental. Unfortunately, the AE
was not able to get the reviewer to elaborate further.

The AE believes that a significant revision of the text is necessary to make the
paper clear and easily understandable, which, in turn, will help to clarify the
significance of the contributions.

**Audience:**

The reviewers largely agree that the problem is important and that the proposed
framework would be of interest to the community. However, a significant rewrite
is necessary for the paper to be most relevant.

**Claims And Evidence:**

The paper presents TopoNet, an end-to-end framework for estimating the topology
of driving scenes, including the relationship between lanes and between lanes
and traffic elements such as traffic lights. Claims regarding the effectiveness
of the method are supported by experimental comparisons to contemporary
baselines along with ablations that highlight the advantages of the proposed
architecture. However, it is difficult to appreciate the significance of these
contributions due to the poor quality of the writing, which requires significant
revisions.

**Resubmission Of Major Revision:**

The authors may consider submitting a major revision at a later time.